# The Role of the Key Effector of Necroptotic Cell Death, MLKL, in Mouse Models of Disease

**DOI:** 10.3390/biom11060803

**Published:** 2021-05-28

**Authors:** Emma C. Tovey Crutchfield, Sarah E. Garnish, Joanne M. Hildebrand

**Affiliations:** 1Department of Medical Education, University of Melbourne, Parkville, VIC 3052, Australia; tovey.e@wehi.edu.au; 2The Walter and Eliza Hall Institute of Medical Research, Parkville, VIC 3052, Australia; garnish.s@wehi.edu.au; 3Department of Medical Biology, University of Melbourne, Parkville, VIC 3052, Australia

**Keywords:** necroptosis, MLKL, programmed cell death

## Abstract

Necroptosis is an inflammatory form of lytic programmed cell death that is thought to have evolved to defend against pathogens. Genetic deletion of the terminal effector protein—MLKL—shows no overt phenotype in the C57BL/6 mouse strain under conventional laboratory housing conditions. Small molecules that inhibit necroptosis by targeting the kinase activity of RIPK1, one of the main upstream conduits to MLKL activation, have shown promise in several murine models of non-infectious disease and in phase II human clinical trials. This has triggered in excess of one billion dollars (USD) in investment into the emerging class of necroptosis blocking drugs, and the potential utility of targeting the terminal effector is being closely scrutinised. Here we review murine models of disease, both genetic deletion and mutation, that investigate the role of MLKL. We summarize a series of examples from several broad disease categories including ischemia reperfusion injury, sterile inflammation, pathogen infection and hematological stress. Elucidating MLKL’s contribution to mouse models of disease is an important first step to identify human indications that stand to benefit most from MLKL-targeted drug therapies.

## 1. Introduction

### Necroptosis and Disease

Mixed Lineage Kinase Domain-Like (MLKL) was shown to be the essential effector of a pro-inflammatory, lytic form of programmed cell death called necroptosis in 2012 [1,2]. Like other forms of lytic cell death (e.g., pyroptosis), necroptosis is characterised by the release of Damage-Associated Molecular Patterns (DAMPs) and IL-33, IL-1α and IL-1β production, as recently reviewed by [3]. Unlike apoptosis, necroptosis and the canonical signalling pathway RIPK1-RIPK3-MLKL are not essential to the development and homeostasis of multicellular organisms [4]. The two most downstream effectors of necroptosis, MLKL and its obligate activating kinase RIPK3, can be deleted at the genetic level in laboratory mice without any overt developmental consequences in the absence of challenge [5,6,7]. Thus, it is widely held that MLKL and necroptosis have evolved primarily to defend against pathogenic insult to cells and tissues. This important role of necroptosis in pathogen defence is written in the DNA of many bacteria and viruses alike, which together encode several genes that disarm different facets of the necroptotic signalling pathway [8,9,10]. In evolution, genetic deletion of MLKL and/or RIPK3 in the ancestors of modern day carnivora, metatheria (marsupials) and aves (birds) show that complex vertebrates can survive without necroptosis when faced with infectious challenges of the ‘real world’ [11]. This adds to the precedent for the well-honed flexibility, redundancy and co-operation of different programmed cell death pathways in the defence against pathogens [12] and offers some biological guide that inhibiting MLKL pharmacologically in humans would not compromise pathogenic defence. Interestingly, recent studies suggest that the suppression of necroptosis may even reduce the resulting inflammatory response that is often more dangerous than the infection itself [13]. *Mlkl* gene knockout (abbreviated to both *Mlkl^−/−^* and *Mlkl* KO) mice are distinguishable from wild-type mice in numerous models of disease, a broad sampling of which are presented (Table 1) and illustrated (Figure 1) here.

The therapeutic potential for necroptosis-targeted drugs lies largely in non-infectious indications. The first human clinical trials of RIPK1-targetted small molecule compounds were conducted in cohorts of rheumatoid arthritis, ulcerative colitis and psoriasis patients [14]. While these showed some promise in phase II, these were returned to the research phase in late 2019 by their licensee GlaxoSmithKline [15]. At the time of writing, there are only three active clinical trials in progress: a phase I trial assessing the safety of a new RIPK1 inhibitor (GFH312), a phase II study utilising RIPK1-binding compound, SAR443122, in cutaneous lupus erythematosus patients and a phase I/II study utilising RIPK1 inhibitor GSK2982772 in psoriasis [16]. There are currently no ‘first in human’ trials of RIPK3- or MLKL-binding compounds listed on http://clinicaltrials.gov (accessed on 28 May 2021).

## 2. *Mlkl* Knock-Out (KO) and Constitutively Active (CA) Mice at Steady State

Following the discovery of its essential role in necroptosis [1,2], two *Mlkl* knockout mouse strains were independently generated by traditional homologous recombination [5] and TALEN technology [7]. More recently, CRISPR-Cas9 engineered *Mlkl^−/−^* [17,33,73], constitutively active point mutant *Mlkl^D139V^* [40], affinity tagged *Mlkl* [67], conditional *Mlkl**^−/−^* strains [29,67] and antisense oligonucleotide (ASO) in vivo *Mlkl* knockdown [60] mouse models have also been used in the study of necroptosis. *Mlkl**^−/−^* mice are born at expected Mendelian ratios and are overtly indistinguishable from wild-type littermates at birth and through to early adulthood [5,7]. Full body histological examination of 2 day old *Mlkl**^−/−^* pups did not reveal any obvious morphological differences, including lesions or evidence of inflammation, relative to wild-type C57BL/6 mice of the same age [40]. Furthermore, no histological differences have been reported for the major organs of young adult *Mlkl**^−/−^* mice [5,7]. Hematopoietic stem cell populations in the bone marrow [5] and CD4/CD8 T cell, B cell, macrophage and neutrophil mature cell populations in secondary lymphoid organs display no observable differences in adult mice [7]. At steady state, serum cytokines and chemokines are indistinguishable from age-matched wild-type littermates [65]. The genetic absence of *Mlkl* and thus necroptosis is generally considered to be innocuous at steady state in the C57BL/6 strain of laboratory mice at a typical experimental age (up to 16 weeks) when housed under conventional clean, pathogen free conditions.

## 3. *Mlkl**^−/−^* Mice in Ischemia and Reperfusion Injury (IRI)

While grouped here for simplicity, MLKL and cell death in general has the potential to influence many facets of the physiological response to blood vessel occlusion/recanalisation and resultant end-organ damage. These facets include the aetiology of the infarction itself, the cellular damage incurred due to the deprivation of oxygen and ATP, the generation of reactive oxygen species that occurs following tissue reperfusion and the inflammation that ensues and convalescence after injury [77]. As summarized in Table 1 (see ‘ischemia and reperfusion injury’), *Mlkl^−/−^* mice appear partially protected from the initial embolic insult [18,21]. For example, *Mlkl^−/−^* mice were reported to exhibit reduced infarct size and have better locomotive recovery day 7 post stroke [18]. This protective effect may in part be due to the role of MLKL and RIPK3 in regulating platelet function and homeostasis [72,78]. *Mlkl* knockout is also protective in models of hepatic and renal IRI [20,79]. Neutrophil activation and inflammation are significant contributors to hepatic IR injury [80]. Despite equivalent levels at steady state, the *Mlkl^−/−^* mice liver parenchyma shows significantly lower numbers of neutrophils 24 h post infarct [17]. This reinforces that the absence of MLKL can play a protective role at the initial ischemic stage and/or at later reperfusion stages, depending on the context.

## 4. Sterile Inflammation

The contribution of MLKL to mouse models of inflammation, which are not borne of pathogenic insult, termed sterile inflammation, is complex and varies according to the initiator, severity, and location within the body. This point is nicely illustrated by systemic inflammatory response syndrome (SIRS). Unlike catalytically inactive RIPK1 or RIPK3 deficiency in mice, *Mlkl**^−/−^* mice are not protected against SIRS driven by *low* dose TNFα [20] or A20 deficiency [20]. When wild-type mice are pre-treated with compound 2, a potent inhibitor of necroptosis that binds to RIPK1, RIPK3 and MLKL, hypothermia is delayed [81]. Together these findings suggest protection against SIRS is necroptosis-independent and occurs upstream of MLKL. *Mlkl^−/−^* mice, however, are significantly protected against SIRS caused by *high* dose TNF [19,20,81]. *Ripk3**^−/−^* mice are similarly protected against high dose TNFα. Remarkably, in contrast to single knockouts, *Ripk3**^−/−^**Mlkl^−/−^* double knockouts resemble wild-type mice and develop severe hypothermia in response to high dose TNFα [19]. This paradoxical reaction is yet to be fully explained by the field. Furthermore, the ablation of MLKL is seen to worsen inflammation induced by non-cleavable caspase-8 (seen in *Casp8^D387A/D387A^* mice) [39] and A20 deficiency [20]. This indicates that necroptosis may serve to limit systemic inflammation in certain scenarios in vivo, a phenomenon also supported by examples of pathogen induced inflammation (see ‘Infection’, Table 1).

One major area of contention is the role of MLKL in mouse models of inflammatory bowel disease [31,32,34,35] and inflammatory arthritis [28,30]; however, key differences in experimental approach may explain these disparities (See Table 1, ‘*Mlkl^−/−^* mice and wild-type control’ details). Similarly, there have been conflicting reports investigating the role of MLKL-driven necroptosis in liver injury [24,25]. Whilst whole body knock-out of *Mlkl* confers protection, independent of immune cells [24], hepatocyte specific ablation of *Mlkl* reveals that necroptosis in parenchymal liver cells is, in fact, dispensable in immune-mediated hepatitis [25]. It would therefore be of interest to know the cell type in which *Mlkl^−/−^* confers protection now that hepatocytes [25] and immune cells [24] have been ruled out. MLKL-deficiency mediates protective effects in models of more localised sterile inflammatory disease: dermatitis [36], cerulein-induced pancreatitis [7], ANCA-driven vasculitis [27], necrotising crescent glomerulonephritis [27] and oxalate nephropathy [22]. Consistent with these findings, mice expressing a constitutively active form of MLKL develop a lethal perinatal syndrome, characterised by acute multifocal inflammation of the head, neck and mediastinum [40].

## 5. Infection: Bacterial

MLKL-dependent necroptosis is thought to have evolved as a pathogen-clearing form of cell death. In support of this theory, 7 of the 10 murine models of bacterial infection examined here led to poorer outcomes in mice lacking MLKL. In response to both acute and chronic infection with *Staphylococcus aureus* and methicillin-resistant *S. aureus* (MRSA), *Mlkl**^−/−^* mice suffer a greater bacterial burden and subsequent mortality [41,42]. This was observed across intravenous, subcutaneous, intraperitoneal, and retro-orbital methods of inoculation [41,42]. Interestingly, despite necroptosis being an inflammatory form of cell death, *Mlkl**^−/−^* mice were found to have greater numbers of circulating neutrophils and raised inflammatory markers (caspase-1 and IL-1β) [41,42]. This is exemplified in models of skin infection where *Mlkl**^−/−^* mice suffer severe skin lesions characterised by excessive inflammation [41]. This suggests that necroptosis is important for limiting bacterial dissemination and modulating the inflammatory response [41]. As an example, MLKL-dependent neutrophil extracellular trap (NET) formation was reported to restrict bacterial replication [42] and contribute to the pathogenesis of inflammatory disease, such as rheumatoid arthritis [82]. In addition to the important infection-busting role of MLKL in neutrophils, non-hematopoietic MLKL is also important for protection against gut-borne infections. MLKL-mediated enterocyte turnover and inflammasome activation were shown to limit early mucosal colonisation by *Salmonella* [49]. MLKL was even shown to bind and inhibit the intracellular replication of *Listeria*, presaging exploration into a more direct, cell-death independent mode of MLKL-mediated pathogen defence [48].

Through co-evolution, bacteria have developed ways to evade, and in some cases, weaponise MLKL and necroptosis for their own ends. Certain bacteria, such as *Serratia marcescens* and *Streptococcus pneumoniae*, produce pore-forming toxins (PFTs) to induce necroptosis in macrophages and lung epithelial cells [47,53]. *Mlkl**^−/−^* mice are consequently resistant to these infections and survive longer than wild-type controls [47,53]. Although PFT-induced necroptosis was reported to exacerbate pulmonary injury in acute infection, it does promote adaptive immunity against colonising pneumococci [46]. *Mlkl**^−/−^* mice demonstrate a diminished immune response, producing less anti-*spn* IgG antibody and thus succumb more readily to secondary lethal *S. pneumoniae* infection [46]. This suggests necroptosis is instrumental in the natural development of immunity to opportunistic PFT-producing bacteria [46]. Interestingly, *Mlkl**^−/−^* has been described to be both protective against [44] and dispensable in [7] the pathogenesis of CLP-induced polymicrobial shock. Results derived from the CLP model of shock can be difficult to replicate, given the inter-facility and even intra-facility heterogeneity of the caecal microbiota in mice [7]. Cell death pathways occur simultaneously during the progression of sepsis, and there is no conclusive evidence of which pathway plays the most deleterious role [7]. Similarly, MLKL appears dispensable for granulomatous inflammation and restriction of *Mycobacterium tuberculosis* colonisation [43].

## 6. Infection: Viral

MLKL can either protect against viral infection or contribute to viral propagation and/morbidity, depending on the type of virus. In initial studies, *Mlkl**^−/−^* mice were indistinguishable from wild-type littermates in response to influenza A (IAV) [13,50,51] and West Nile virus infection [54]. Recent findings, however, suggest that *Mlkl* knockout confers protection against lung damage from lethal doses of influenza A [13]. Despite equivalent pulmonary viral titres to wild-type littermates, *Mlkl**^−/−^* mice had a considerable attenuation in the degree of neutrophil infiltration (~50%) and subsequent NET formation and thus were protected from the exaggerated inflammatory response that occurs later in the infection [13]. In line with this finding, *Mlkl**^−/−^* mice are protected against bacterial infection secondary to IAV [53]. A recent study also finds that MLKL-deficiency mediates protection against cardiac remodelling during convalescence following IAV infection by upregulating antioxidant activity and mitochondrial function [52], indicating the potential utility of MLKL-targeted therapies for both the acute and long term effects of viral infection.

## 7. Metabolic Disease

MLKL deficiency has shown diverse effects in at least four separate studies of non-alcoholic fatty liver disease (NAFLD). After 18 hours of a choline-deficient methionine-supplemented diet [55] or 8 weeks of a Western diet [17], *Mlkl**^−/−^* mice are indistinguishable from wild-type controls. Following 12 weeks of a high fat diet (HFD) [56], however, *Mlkl**^−/−^* mice appear resistant to steatohepatitis given that MLKL-deficiency promotes reduced de novo fat synthesis and chemokine ligand expression [56]. A similar effect is seen following 12 weeks of a high fat, fructose, and cholesterol diet, where *Mlkl**^−/−^* mice are markedly protected against liver injury, hepatic inflammation and apoptosis attributed to inhibition of hepatic autophagy [57]. In line with these findings, mice treated with RIPA-56 (an inhibitor of RIPK1) downregulate MLKL expression and were found to be protected against HFD-induced steatosis [83]. These findings offer a tantalising clue that necroptosis may contribute to this disease, depending on the trigger. In contrast to NAFLD, MLKL does not appear to play a statistically significant role in acute or chronic alcoholic liver disease [58].

MLKL deficiency provides variable protection against the metabolic syndrome, depending on the challenge. MLKL deficiency appears to protect against dyslipidaemia with reduced serum triglyceride and cholesterol levels following a high fat diet [56] and Western diet [60], respectively. Whilst *Mlkl**^−/−^* mice appear to have significantly improved fasting blood glucose levels given improved insulin sensitivity following 16 weeks of a high fat diet [59], there is conflicting evidence on the effect at steady state [56,59]. There is also conflicting evidence on the role of MLKL deficiency in adipose tissue deposition and weight gain [56,59]. Whilst comparable at baseline, after 16 weeks of HFD, *Mlkl**^−/−^* mice gained significantly less body weight, notably visceral adipose tissue, than their wild-type littermates [59]. Blocking upstream RIPK1, has a similar effect [84]. Yet, in another study, after 12 weeks of HFD there were no significant differences found in body weight between *Mlkl**^−/−^* and wild-type [56]. Finally, MLKL has been shown to play a role in atherogenesis; MLKL facilitates lipid handling in macrophages, and upon inhibition, the size of the necrotic core in the plaque is reduced [60]. Of the eight broad disease classes covered in this review, the role of MLKL in metabolic disease is arguably the most disputed, owing to the long-term nature of the experiments and the propensity for confounding variables, including genetic background and inter-facility variation in microbiome composition. The field may benefit from a more standardised approach to metabolic challenge and the prioritization of data generated using congenic littermate controls.

## 8. Neuromuscular

Evidence is rapidly accumulating for the role of MLKL in mouse models of neurological disease. *Mlkl**^−/−^* mice were reported to be protected in one model of chemically-induced Parkinson’s disease, with a significantly attenuated neurotoxic inflammatory response contributing to higher dopamine levels [69]. Strikingly, recent evidence suggests that MLKL may be important for tissue regeneration following acute neuromuscular injury [67,70]. In a model of cardiotoxin-induced muscle injury, muscle regeneration is driven by necroptotic muscle fibres releasing factors into the muscle stem cell microenvironment [70]. *Mlkl**^−/−^* mice accumulate massive death-resistant myofibrils at the injury site [70]. Furthermore, in a model of sciatic nerve injury, MLKL was reported as highly expressed by myelin sheath cells to promote breakdown and subsequent nerve regeneration [67]. Overexpression of MLKL in this model is found to accelerate nerve regeneration [67] speaking to the potential of MLKL enhancing rather than blocking drugs in this area. However, contraindicating the use of MLKL activating drugs to mitigate neurological disease is the observation that MLKL accelerates demyelination in a necroptosis-independent fashion and thereby worsens multiple sclerosis pathology [68]. Finally, the role of necroptosis in murine amyotrophic lateral sclerosis remains contentious in the field. Wang et al. (2020) reported that MLKL-dependent necroptosis appears dispensable in the onset, progression, and survival of SOD1^G93A^ mice. Yet, there have been robust studies suggesting RIPK1-RIPK3-MLKL drives axonal pathology in both SOD1^G93A^ and *Optn**^−/−^* mice [85,86]. While opinions in the field remain split on the relative contribution of MLKL in hematological (which express high levels of MLKL) vs. non-hematological (i.e., neurons express low levels of MLKL at baseline) cells in many of these models, disorders of the neuromuscular system have clearly come to the fore in commercial necroptosis drug development efforts.

## 9. Hematological

*Mlkl**^−/−^* mice are hematologically indistinguishable from wild-type at steady state [5,79,87]. This trend continues as the mice age to 100 days [65]. Properly regulated necroptosis, however, is indispensable for hematological homeostasis. *Mlkl^D139V/D139V^* mice (which encode a constitutively active form of MLKL that functions independently of upstream activation from RIPK3) have severe deficits in platelet, lymphocyte, and hematopoietic stem cell counts [40]. Mice expressing even one copy of this *Mlkl^D139V^* allele are unable to effectively reconstitute the hematopoietic system following sub-lethal irradiation or in competitive reconstitution studies [40]. Like *Ripk3**^−/−^* mice, *Mlkl**^−/−^* mice display a prolonged bleeding time and thus unstable thrombus formation [72,78]. MLKL, however, appears to play an additional, RIPK3-independent, role in platelet formation/clearance. In a model of lymphoproliferative disorder, *Casp8**^−/−^**Mlkl**^−/−^* double knockout mice develop a severe thrombocytopaenia that worsens with age (measured at 50 and 100 days) [65]. This phenotype is not observed in age-matched *Casp8**^−/−^**Ripk3**^−/−^* mice [65]. Finally, MLKL has also been shown to function in neutrophil NET-formation at the cellular level [42], and *Mlkl**^−/−^* mice are seen to be protected from diseases that implicate NET-formation, for example ANCA associated vasculitis [27] and deep vein thrombosis [71].

## 10. Cancer and Cancer Treatment

Cisplatin is a common chemotherapy that treats solid cancers, although its utility is limited by its nephrotoxic effects [23]. Strikingly, *Mlkl**^−/−^* mice are reported to be largely resistant to cisplatin-induced tubular necrosis compared to wild-type controls [23]. However, mechanistically it is still not understood how systemic delivery of a DNA-damaging agent such as cisplatin could induce renal necroptosis and whether it is particular to kidney tissue. Hematopoietic stem cells derived from *Mlkl**^−/−^* mice play a key role in studies showing that apoptosis-resistant acute myeloid leukemia (AML) could be forced to die via necroptosis [61]. By adding the caspase inhibitor IDN-6556/emricasan, AML cells are sensitised to undergo necroptosis in response to the known clinical inducer of apoptosis SMAC mimetic, birinapant [61]. IDN-6556/emricasan is well-tolerated in humans [88] and is an excellent example of a therapeutic approach designed to enhance rather than block MLKL activity in vivo. Finally, the role of MLKL in intestinal tumorigenesis remains unclear, with studies reporting that MLKL is dispensable in both sporadic intestinal or colitis-associated cancer [35], and yet MLKL has been reported to have a protective effect in *Apc^min/+^* mice [62] by suppressing IL-6/JAK2/STAT3 signals [63].

## 11. Reproductive System

There are two studies that assess the role of MLKL in age-induced male infertility. *Mlkl**^−/−^* mice aged to 15 months demonstrate significantly reduced body, testicular and seminal vesicle weight alongside increased testosterone levels and fertility [73] when compared to age matched wild-type mice. A recent follow up study suggests that CSNK1G2, a member of the casein kinase family, is co-expressed in the testes and inhibits necroptosis-mediated aging [89]. This phenomenon is also seen in human testes [89]. Another study that uses congenic littermate controls, however, finds that male mice aged to 18 months are indistinguishable from wild-type littermates with regards to total body, testicular and seminal vesicle weight [74].

## 12. Important Experimental Determinants in MLKL-Related Mouse Research

There are several high profile examples of genetic drift [90], passenger mutations [90], facility-dependent variation in the microbiomes of mice [91], sex [92] and age [93] acting as important modifiers of the innate immune response. One notable example is the significant differences between commonly used wild-type control C567BL/6J and C57BL/6NJ (also known as C57BL/6N) sub-strains (separated by 60+ years of independent breeding) in morbidity and survival following LPS- and TNFα-induced lethal shock [20,90]. Of similar interest, male mice are reported to be more susceptible than females to invasive pneumonia and sepsis [94]. Generously powered cohorts of sex-segregated, co-housed, congenic littermates (mice derived from a heterozygous cross) are the gold standard for controlling these confounding variables when comparing wild-type and *Mlkl* KO/mutant mice (or any other mutant) [91]. All scientists that work with mice will attest that while this approach is certainly time, resource, and mouse-number intensive, it should nonetheless be prioritised by experimenters and peer reviewers alike wherever possible. We congratulate the efforts of scientists who proceed one step further by restoring the wild-type phenotype through ectopic expression of MLKL [67]. When the use of littermate controls is not feasible, clear, and detailed descriptions of mouse age, sex and provenance provided in publications act as important caveats for discerning reviewers and readers to consider. In Table 1, we summarize the outcomes of these comparisons in 80+ contexts and have included a column that provides details of caveats where available.

## 13. Concluding Remarks

Genetic and experimental models of disease provide a strong rationale that MLKL and necroptosis are important mediators and modifiers of infectious and non-infectious disease. The study of MLKL in mouse models has indicated a bidirectional role of necroptosis in disease. *Inhibiting* the function of MLKL may confer protection in diseases characterised by hematopoietic dysfunction and uncontrolled inflammation borne of (and/or perpetuating) the failure of epithelial barriers throughout the body. *Enhancing* MLKL-induced cell death may prove beneficial in the treatment of malignancies and nerve injury. The therapeutic potential of MLKL as a druggable target in infectious disease is highly nuanced and will require careful tailoring to the pathogen and infection site/stage in question. Importantly, any extrapolation of these observations in *Mlkl* knockout and mutant mice to human disease must be tempered by our knowledge of key differences in both the structure and regulation of mouse and human MLKL [95,96,97]. This point is particularly poignant considering withdrawals of certain RIPK1 inhibitor drugs from phase I (pancreatic cancer) and II (chronic inflammatory diseases) clinical trials due to lack of efficacy [98,99]. Rapid advances in the ‘humanisation’ of mouse models of disease and the use of large human clinico-genetic databases will further enrich the extensive body of murine data supporting the role of MLKL in human disease.

## Figures and Tables

**Figure 1 biomolecules-11-00803-f001:**
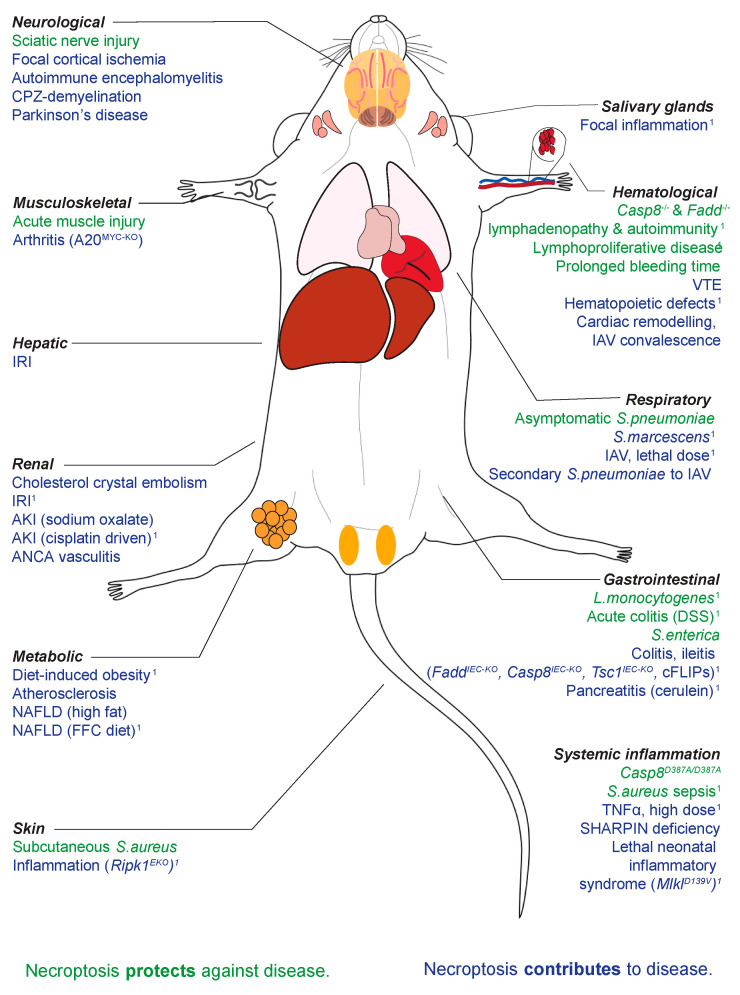
The role of necroptosis in infective and non-infective challenges spanning a wide array of physiological systems. ^1^ Littermate controls utilised.

**Table 1 biomolecules-11-00803-t001:** Understanding the role of necroptosis in the aetiology of disease using *Mlkl^−/−^* and mutant mice.

Challenge	Method	Outcome in *Mlkl^−/−^* (Mutant or Knock-Down) Mouse Relative to Wild-Type Control	*Mlkl^−/−^* Mouse and Wild-Type Control Details	Reference
Ischemia and reperfusion injury (IRI)				
Hepatic IRI	45 min ischemia in chow or Western diet fed mice.	Reduced TUNEL positive hepatocytes and serum ALT levels in *Mlkl^−/−^* mice, irrespective of diet. Hepatic neutrophil infiltration and TNF, IL-1, MIP-2, and IL-6 mRNA reduced post I/R in *Mlkl^−/−^* mice.	Sex-matched *Mlkl^−/−^* (CRISPR-Cas9) and *Wt* mice utilised.	[17]
Renal infarction, no reperfusion	Cholesterol crystal embolism.	*Mlkl^−/−^* mice protected from infarction (measured by infarct size, kidney injury and neutrophil infiltration). No difference in the extent of acute kidney injury (measured by eGFR) and kidney failure.	*Mlkl^−/−^* [5] and *Wt* mice utilised.	[18]
Renal IRI	40 min bilateral renal pedicle clamp.	*Mlkl^−/−^* mice protected from acute kidney injury (measured using serum creatinine and urea). *Ripk3^−/−^Mlkl^−/−^* double knockout mice were only somewhat protected.	Sex- and weight-matched *Wt* controls for *Mlkl^−/−^* [5] mice.	[19]
	30 min renal pedicle clamp.	*Mlkl^−/−^* prolonged survival by ~4 days over *Wt* mice.	Male *Wt* and *Mlkl^−/−^* [5] littermates.	[20]
Focal cortical ischemia	Photothrombosis of the cortical microvessels.	Reduced infarction (fewer TUNEL/NeuN-positive cells) in *Mlkl^−/−^* mice and improved locomotion from day 7. Fewer F4/F80+ cells and reduced expression of iNOS, TNFα, IL-12 and IL-18.	*Mlkl^−/−^* sourced from Xiamen University *	[21]
Sterile Inflammatory				
Acute kidney injury (AKI)	Intraperitoneal injection of sodium oxalate.	*Mlkl^−/−^* mice show reduced serum creatinine, neutrophil recruitment, and tubular necrosis (TUNEL staining and tubular injury score).	Sex matched male *Wt* controls for *Mlkl^−/−^* [5] mice.	[22]
	Intraperitoneal injection of cisplatin.	*Mlkl^−/−^* mice resistant to tubular necrosis (histology, blood urea nitrogen and serum creatinine). Reduced TNFα, IL-1β, IFN-γ, IL-6 in proximal tubules despite similar levels to *Wt* at baseline.	Male *Wt* and *Mlkl^−/−^* (TALEN) mice are littermates.	[23]
Hepatitis	ConA-induced.	*Mlkl^−/−^* mice show reduced hepatocyte necrosis (TUNEL staining and ALT/AST).	*Wt* and *Mlkl^−/−^* [5] a combination of C57BL/6J or littermate controls used.	[24]
		Hepatocyte necrosis in *Mlkl*^LPC-KO^ mice, with specifically ablated MLKL in liver parenchymal cells, are indistinguishable from *Wt* (ALT and H&E).	*Mlkl*^LPC-KO^ and *Mlkl^fl/fl^* are littermates.	[25]
	LPS/GalN-driven (model of apoptotic hepatitis).	*Mlkl^−/−^* mice are indistinguishable from *Wt* (assessed using ALT/AST and histological staining for cleaved Casp3).	*Wt* and *Mlkl^−/−^* [5] a combination of C57BL/6J or littermate controls used.	[24]
	Acetaminophen (model of drug-induced liver injury).	MLKL deficiency does not prevent liver injury (assessed using AST/ALT levels and TUNEL staining).	*Wt* and *Mlkl^−/−^* [5] a combination of C57BL/6J or littermate controls used.	[24]
			Male *Wt* used for *Mlkl^−/−^* [5] mice. Bred as separate cohorts.	[26]
Acute pancreatitis	Cerulein-induced.	*Mlkl^−/−^* mice experience less severe acinar cell necrosis than *Wt* mice (as measured by histological and quantitative analysis).	Male *Mlkl^−/−^* (TALEN) and *Wt* mice are littermates.	[7]
ANCA-driven necrotising crescent glomerulonephritis	αMPO IgG transfer.	*Mlkl^−/−^* mice were protected against NCGN (as measured by reduced leukocyturia, erythrocyturia and less crescents and necrotic change on histology).	*Mlkl^−/−^* [5]. *	[27]
Arthritis	K/B X N serum transfer.	No significant difference in disease progression between *Mlkl^−/−^* and *Wt* mice (clinical severity scale and myeloperoxidase average radiance).	Sex matched *Wt* for *Mlkl^−/−^* mice.	[28]
	Myeloid-cell-specific A20 deficiency (A20^MYC-KO^).	A20^MYC-KO^*Mlkl^−/−^* mice were protected against inflammatory arthritis (thickness of rear ankles, histological scores of inflammation, cartilage and bone destruction), splenomegaly and showed reduced expression of IL-1β and TNFα.	Mice with loxP-flanked A20 crossed with *Mlkl^−/−^* [29] aged to 23-to-27 weeks old.	[30]
Colitis and ileitis	IEC-specific FADD and Caspase-8 deficiency.	*Mlkl^−/−^Fadd^IEC-KO^* and *Mlkl^−/−^Casp8^IEC-KO^* resistant to colitis. *Mlkl^−/−^Casp8^IEC-KO^* are also resistant to ileitis whilst *Mlkl^−/−^Fadd^IEC-KO^* are only partially protected. Assessed using histological analysis of colonic and ileal tissues and microarray.	Littermate control mice with homozygous or heterozygous loxP-flanked Fadd or Casp8.	[31]
	IEC-specific highly active mTORC1 pathway (*Tsc1^IEC-KO^*).	*Tsc1^IEC-KO^Mlkl^−/−^* profoundly alleviates epithelial cell death and intestinal barrier dysfunction that is seen in *Tsc1^IEC-KO^.*	*Tsc1^IEC-KO^* and *Tsc1^IEC-KO^Mlkl^−/−^* were age- and sex-matched littermate controls.	[32]
Lethal ileitis	X-linked transgene -short form of cellular FLICE-inhibitory protein (cFLIPs).	MLKL deficiency partially rescues male mice from lethal in-utero ileitis (improved survival, small intestine villous architecture, fewer CC3-positive cells).	Male *Wt* and *Mlkl^−/−^* [33].	[34]
Acute colitis	1.5% DSS in drinking water, 5 days, followed by 3 days of normal drinking water.	*Mlkl^−/−^* mice display increased weight loss, but only in some cohorts. Day 5 and 8, indistinguishable murine endoscopic index of colitis (MEIC) scores. Increased inflammation in the proximal colon and decreased submucosal inflammation in the distal colon (H&E).	*Wt* and *Mlkl^−/−^* [5] are littermates.	[35]
Inflammatory skin disease	RIPK1 deficient keratinocytes *Ripk1^EKO^.*	Specific MLKL deficiency in keratinocytes profoundly ablated the development of the inflammatory skin lesion that invariably develops in *Ripk1^EKO^* mice.	*Ripk1^FL/FL^* (Takahashi et al., 2014) *Mlkl^FL/FL^* (Murphy et al., 2013) *Wt* littermates utilised.	[36]
Systemic inflammatory response syndrome (SIRS)	300 μg/kg TNF intravenous/kg TND.	*Mlkl^−/−^* mice suffer a hypothermia reaction like that of *Wt* controls.	*Wt* and *Mlkl^−/−^* [5] are littermates.	[20]
	500 μg/kg TNF intravenous.	*Mlkl^−/−^* mouse core body temperature higher at 6, 8 and 10 h post dose than *Wt* controls, indicating moderate protection.	*Wt* and *Mlkl^−/−^* [5] are littermates.	[20]
	30 mg/kg LPS-intraperitoneal.	*Mlkl^−/−^* mice show similar serum TNFα and IL-1β levels as *Wt* (measured at 0, 2, 4 and 8 h by ELISA).	Male *Mlkl^−/−^* (TALEN) and *Wt* mice are littermates.	[7]
	1 mg/kg TNF in 200 μL PBS via the tail vein.	*Mlkl^−/−^* mice showed increased survival. *Ripk3^−/−^Mlkl^−/−^* double knockout mice were not similarly protected.	Female *Wt* and *Mlkl^−/−^* [5] mice bred as separate cohorts.	[19]
	50 μg/kg LPS intraperitoneal.	*Mlkl^−/−^* mice showed comparable levels of TNFα and IL-6 (ELISA analysis) 1 h post-treatment. Pre-treatment with necrostatin-1 (IV), 15 min prior to LPS, significantly reduced *Mlkl^−/−^* mice ability to produce TNFα and IL-6.	*	[37]
Generalised inflammation	A20 gene deficiency.	*Mlkl^−/−^A20^−/−^* mice display similar inflammation (hepatic neutrophil infiltration, RANTES secretion and dermatitis) and lifespan.	*Wt* are *A20^−/−^* littermates.	[20]
TNF-induced multi-organ inflammation	SHARPIN deficiency *shpn^m/m^.*	12-week-old *Shpn^m/m^Mlkl^−/−^* mice show reduced liver inflammation, splenomegaly and leucocytosis in comparison to *Shp^m/m^* controls (histology, ADVIA automated hematological analysis).	C57BL/Ka *Shpn^cpdm/cpdm^* backcrossed to C57BL/6J one or two times or crossed with C56BL/6J *Mlkl^−/−^* [5] mice.	[38]
*Casp8^D387A/D387A^* (non-cleavable caspase 8)-induced systemic inflammation	*Casp8^D387A/D387A^* mutation.	*Mlkl^−/−^ Casp8^DA/DA^* do not develop LPR disease. *Mlkl^−/−^ Casp8^DA/DA^* show an exacerbated inflammatory phenotype with significant splenomegaly, liver damage (serum ALT, AST) and premature death.	*Casp8^D387A/D387A^* (CRISPR-Cas9) and *Mlkl^−/−^* [5] mice are sex-matched.	[39]
*Mlkl^D139V^*-induced lethal neonatal inflammatory syndrome	Whole body homozygosity for constitutively active *Mlkl^D139V^* mutation.	*Mlkl^D139V^* homozygotes are born normal but develop acute multifocal inflammation of the head, neck, and mediastinum by day P2/P3. Maximum lifespan observed 6 days.	Age and sex matched littermate controls.	[40]
Infection				
Bacterial				
*Staphylococcus aureus*	Subcutaneous 2 × 10^6^ CFU of *S. aureus* strain MRSA USA300.	5-day p.i. *Mlkl^−/−^* mice suffer greater bacterial burdens, larger skin lesions, significantly increased neutrophil, macrophage, *γδ* T cell infiltrate and enhanced pro-inflammatory cytokine expression.	Sex-matched *Wt* and *Mlkl^−/−^* [5].	[41]
	Retro-orbital 1 × 10^8^ CFU *S*. *aureus* strain MRSA USA300.	*Mlkl^−/−^* mice show increased mortality rates compared to WT (a mean survival of 4 days).	Sex-matched *Wt* and *Mlkl^−/−^* [5].	[41]
	Intravenous 1 × 10^7^ CFU of *S. aureus* strain MRSA.	Accelerated weight loss and morbidity in *Mlkl^−/−^* mice. Circulating neutrophil numbers elevated 24 h p.i. Increased CFU in the blood and MRSA burden in the kidney of *Mlkl^−/−^* mice.	*Mlkl^−/−^* [5] mice littermates utilised.	[42]
	Retro-orbital 1 × 10^6–7^ CFU of *S. aureus* strain MRSA.	*Mlkl^−/−^* mice show increased MRSA burden in blood and kidneys and greater number of peripheral neutrophils (flow cytometry).	*Mlkl^−/−^* [5] mice littermates utilised.	[42]
	Intraperitoneal 1 × 10^7^ CFU of *S. aureus* strain MRSA.	*Mlkl^−/−^* mice display more severe bacteraemia 24 h p.i.	*Mlkl^−/−^* [5] mice littermates utilised.	[42]
*Mycobacterium tuberculosis*	~100–200 CFU of *Mtb* via aerosol strain H37Rv.	*Mlkl^−/−^* mice are indistinguishable from *Wt* controls in terms of splenic and respiratory bacterial burden, gross lung histopathology of inflammatory lesions (number and size), organisation of granulomatous inflammation, immune cell counts, TNFα and IL-1β.	Sex-matched *Wt* and *Mlkl^−/−^* [5] mice.	[43]
Polymicrobial septic shock	Cecal ligation and puncture (CLP).	*Mlkl^−/−^* and *Wt* mice showed same survival profile (mortality was monitored from 24 h to 144 h).	Age- and sex-matched littermates of *Wt* and *Mlkl^−/−^.*	[7]
		*Mlkl^−/−^* have better survival (~50%) than *Wt* controls (~25%) at 180 h and are moderately protected against hypothermia. *Mlkl^−/−^* experience less severe lung, small intestine (H&E) injury with reduced levels of ALT, serum BUN, HMGB1, TF, IL-1β, TNFα.	Age- and sex-matched *Wt* and *Mlkl^−/−^*.	[44]
Acute kidney injury (AKI) following polymicrobial septic shock	Cecal ligation and puncture (CLP).	*Mlkl^−/−^* mice suffer same extent of AKI. Ratio of lipocalin-2/urine creatinine levels were lower in *Mlkl^−/−^* mice than *Wt*, however not as reduced as *Ripk3^−/−^* mice.	*Wt* and *Mlkl^−/−^* (Jiahuai Han laboratory) littermates.	[45]
Asymptomatic chronic nasopharyngeal colonisation with *S. pneumoniae*	Nasal instillation of ~1 × 10^5^ CFU of serotype 4 strain TIGR4 m.	*Mlkl^−/−^* mice demonstrate reduced nasopharyngeal epithelial cell sloughing and increased LDH, Il-33, IL-1α, CXCL2 levels and decreased Il-6, Il-17 and polymorphonuclear cells than *Wt*. *Mlkl^−/−^* mice cleared *S. pneumoniae* colonisation at a slower rate than *Wt* controls. Less anti-PspA IgG than *Wt* controls despite comparable total serum IgG concentration.	*Wt* and *Mlkl^−/−^* [5] mice utilised.	[46]
*S. marcescens* hemorrhagic pneumonia	Intratracheal infection with strain MB383.	*Mlkl^−/−^* mice have increased alveolar macrophages and suffer less lung damage (histological analysis).	*Wt* and *Mlkl^−/−^* littermates (Douglas Green).	[47]
*Listeria monocytogenes*	Oral 1 × 10^8^ CFU *L. monocytogenes* 1/2b strain 2011L-2858.	3 days p.i. *Mlkl^−/−^* mice have a moderate increase in liver bacterial colonisation compared to *Wt* littermate controls.	Female *Wt* and *Mlkl^−/−^* [5] littermates.	[48]
*Salmonella enterica*	Oral 5 × 10^7^ CFU −1 × 10^8^ CFU *Subsp. Enterica serovar Typhimurium* strain SL1344.	*Mlkl^−/−^* mice show greater submucosal oedema, loss of goblet cells, PMN infiltration, loss of epithelial barrier integrity and salmonella colonisation despite comparable bacterial faecal loads. Also have increased body and cecal weight loss.	*Wt* and *Mlkl^−/−^* (Dr. Jia-Huai Han, Xiamen University, China) were sex-matched.	[49]
Viral				
Influenza A strain PR8	Instranasal 4000 EID50.	*Mlkl^−/−^* mice indistinguishable from *Wt* in terms of survival (75% of both *Mlkl^−/−^* and *Wt* mice survived and recovered) and lung progeny virion output.	Sex-matched *Wt* controls *.	[50]
	Instranasal 2500 EID_50_. Intranasal 6000 EID_50_.	At moderate dose, *Mlkl^−/−^* mice are indistinguishable from *Wt* in terms of survival. At lethal dose, *Mlkl^−/−^* mice show increased survival; fewer disrupted epithelial cells despite same viral titre 6 days p.i. Influx of neutrophils into the lungs in *Mlkl^−/−^* mice both delayed and diminished.	Sex-matched littermates used for *Mlkl^−/−^* mice [5].	[13]
	Instranasal 2500 EID_50_. Instranasal 1500 EID_50_.	*Mlkl^−/−^* are indistinguishable from *Wt* controls in terms of survival, viral titres, morphometry of viral spread and percentage of infected lung. IAV-mediated alveolar inflammation; septal thickening, inflamed alveoli, and hyaline membranes in *Mlkl^−/−^* mice comparable to that of *Wt* mice.	Sex matched *Wt* littermate or *Wt* C57BL/6 mice were used as controls for *Mlkl^−/−^* mice [5].	[51]
Influenza strain A/California/7/2009	250 PFU of IAV.	At baseline, *Mlkl^−/−^* mouse myocardium showed increased mitochondrial and antioxidant activity (proteome analysis), increased survival and reduced weight loss.	*Wt* (B6NTac) mice and *Mlkl^−/−^* mice [5] were bred independently.	[52]
Secondary *S. pneumoniae* following influenza A viral infection	Intranasal 250 PFU pdmH1N1. Intratracheal 1 × 10^3^ CFU of *S. pneumoniae*.	*Mlkl^−/−^* mice showed reduced pulmonary cell death (TUNEL-staining), bacterial burden, lung consolidation and IFN-α, -β expression. No changes in the amount of oxidative-stress-induced DNA damage (immunofluorescence of 8-OHdG).	*Wt* and *Mlkl^−/−^* [5].	[53]
West Nile Virus (WNV) encephalitis	Subcutaneous injection with 100 pfu of WNV-TX 2002-HC strain.	*Mlkl^−/−^* mice are indistinguishable from *Wt* in terms of survival and viral titres.	Sex matched *Wt* C57BL/6J used for *Mlkl^−/−^* mice [5].	[54]
Metabolic				
Non-alcoholic fatty liver disease	Choline-deficient, methionine-supplemented (CDE), fed once.	*Mlkl^−/−^* mice were indistinguishable from *Wt* controls in terms of hepatic necrosis (serum AST and PI-positive cells). Reduced systemic levels of Il-6 and IL-1β (RT-PCR), comparable levels of TNFα.	Male C57BL/6J (CLEA Japan) including *Mlkl^−/−^* (M.Pasparakis) were utilised.	[55]
	12 weeks of high fat diet.	*Mlkl^−/−^* mice gain body weight comparable to *Wt*. At baseline, *Mlkl^−/−^* mice had increased serum AST/ALT and decreased serum fasting blood glucose. *Mlkl^−/−^* mice demonstrate reduced NAFLD activity score, steatosis score, hepatocyte ballooning, lobular inflammation, serum AST/ALT, triglyceride levels and de novo fat synthesis.	C57BL/6N *Wt* and *Mlkl^−/−^* (Jiahuai Han) mice utilised.	[56]
	8 weeks of Western diet.	*Mlkl^−/−^* are indistinguishable from *Wt* in terms of level of steatosis and liver triglyceride accumulation.	Sex-matched *Mlkl^−/−^* (CRISPR-Cas9) on C57BL/6N and *Wt* utilised.	[17]
	12 weeks of fat, fructose and cholesterol (FFC) diet.	*Mlkl^−/−^* mice are protected from liver injury (measured using AST/ALT, hepatic triglyceride accumulation, macrovesicular and microvesicular steatosis (H&E). *Mlkl^−/−^* mice were protected from FFC-induced apoptosis (measured using M30 levels, cleaved caspase-3, and TUNEL-positive cells) and inflammation (measured using mRNA levels of TNFα, Il-1β, MCP-1 and F4/80).	Littermates utilised.	[57]
Alcoholic fatty liver disease (ALFD)	Chronic ethanol-induced.	*Mlkl^−/−^* mice are indistinguishable from *Wt* in terms of ALT/AST, hepatic triglycerides, macrovesicular and microvesicular steatosis (H&E).	Sex-matched littermates utilised.	[58]
	Gao-binge.	*Mlkl^−/−^* mice are indistinguishable from *Wt* in terms of body weight, food intake, ALT/AST, hepatic triglycerides, macrovesicular and microvesicular steatosis (H&E). *Mlkl^−/−^* mice have similar levels of CYP2E1, ER stress and hepatocyte apoptosis but mildly reduced levels of some hepatic inflammatory markers.	Sex-matched littermates utilised.	[58]
Diet-induced obesity	16 weeks of high fat diet (HFD) consisting of 60% kcal from fat or chow diet (CD).	*Mlkl^−/−^* mice on regular CD are indistinguishable from *Wt* littermates in terms of body weight, glucose disposal, glucose tolerance, or insulin sensitivity. After 16 weeks on HFD, *Mlkl^−/−^* mice have lower body weight and visceral adipose, and better glucose and insulin tolerance. No difference in inflammatory markers or TUNEL positive cells in the liver.	Body weight matched, male *Wt* and *Mlkl^−/−^* littermates are utilised.	[59]
Atherosclerosis	Western diet for 8 or 16 weeks.	*Apoe^−/−^* mice fed a Western diet while receiving MLKL antisense oligonucleotides (ASOs) demonstrated reduced necrotic core size of aortic sinus plaques, reduced plasma cholesterol, fewer TUNEL positive cells but increased lipid content in atherosclerotic plaque (oil red O staining).	*Apoe^−/−^* C57Bl/6N mice administered with control or MLKL antisense oligonucleotides (ASOs).	[60]
Neoplasia				
Acute myeloid leukemia	Retroviral expression of the fusion protein (MLL-ENL) in *Mlkl^−/−^* hematopoietic stem cells.	AML cells (MLL-ENL-transduced E14 liver hematopoietic stem cells (HSCs)) transplanted into lethally irradiated *Wt* mice. AML generated from *Mlkl^−/−^* HSCs showed similar leukemia progression and overall survival compared to AML generated from *Wt* HSCs.	Age matched, both *Mlkl^−/−^* and *Wt* on C57B/6J background.	[61]
Colon cancer	Sporadic intestinal adenoma (APC^min^ mouse).	No significant difference in colonic tumour burden. No difference in colonic inflammation (H&E) or expression of Il-6.	*Wt* and *Mlkl^−/−^* [5] are littermates. Data for both sexes presented.	[35]
	Colitis-associated cancer (azoxymethane, DSS).	No significant difference in weight loss observed. Similar timing of tumour onset and burden between *Mlkl^−/−^* and *Wt* (endoscopic tumour scores and H&E).	*Wt* and *Mlkl^−/−^* [5] are littermates. Data for both sexes presented.	[35]
Colitis-associated tumorigeneses	AOM injection at 10 mg/kg of body weight. Five days post, fed with three cycles of 3% (*w*/*v*) DSS, followed by 14 days of normal water.	*Mlkl^−/−^* mice have a significant reduction in body weight, increased clinical severity, shorter colons, and worse survival in comparison to *Wt*. *Mlkl^−/−^* mice exhibit increased burden of anal and colonic polyps with significant increase in inflammation, hyperplasia, and dysplasia (H&E).	*Wt* and *Mlkl^−/−^* (CRISPR-Cas9) were sex-matched (males only).	[62]
		*Apc^min/+^ Mlkl^−/−^* mice have a median survival time of 127 days compared with *Apc^min/+^* survival of 185 days, alongside marked increases in tumour number and load.	*Wt* and *Mlkl^−/−^* (CRISPR-Cas9) were littermate controls.	[63]
Progressive lymphoproliferative disease	*Fadd* gene knock out.	*Mlkl^−/−^* rescues *Fadd^−/−^* mice from embryonic lethality but results in more severe progressive lymphoproliferative disease compared to *Ripk3^−/−^Fadd^−/−^* mice.	*Mlkl^−/−^* generated by CRISPR-Cas9 (Bioray Labs).	[64]
			Littermates derived from *Fadd^+/−^, Mlkl^+/−^* crosses [5] mice utilised.	[65]
	*Casp8* gene knock out.	*Mlkl^−/−^* rescues *Casp8^−/−^* mice from embryonic lethality but results in more severe progressive lymphoproliferative disease compared to *Ripk3^−/−^Casp8^−/−^* mice.	Littermates derived from *Casp8^+/−^, Mlkl^+/−^* crosses [5] mice utilised.	[65]
Neuromuscular				
Amyotrophic lateral sclerosis	*SOD1^G93A^* mice (express mutant human superoxide dismutase 1).	MLKL deficiency does not affect disease onset, progression, or survival in *SOD1^G93A^* mice. MLKL ablation has no impact on astrocyte or microglial activation.	Four isogenic genotypes for study: *SOD1^G93A^;**Mlkl^−/−^*, *SOD1^G93A^, Mlkl^−/−^* and *Wt* littermates. *Mlkl^−/−^* mice [5].	[66]
Sciatic nerve crush injury	Sciatic nerve cut or crushed unilaterally.	pMLKL (serine 441) is upregulated in damaged sciatic nerve cells of both *Wt* and *Ripk3^−/−^* mouse sciatic nerves. *Mlkl^−/−^* mice suffer drastically reduced myelin sheath breakdown and thus reduced sciatic nerve regeneration and function.	Male only *Mlkl^−/−^* (Li et al., 2017) and *Wt* (C57BL/6J) utilised.	[67]
Experimental autoimmune encephalomyelitis (EAE)	Intraperitoneal injection of 200ng of pertussis toxin.	*Mlkl^−/−^* mice display significantly reduced clinical EAE disease scores with delayed reduction of myelination (MBP immunofluorescence).	Sex matched *Wt* and *Mlkl^−/−^* (CRISPR-Cas9) bred as separate cohorts. Same housing conditions since birth.	[68]
Cuprizone CPZ-induced demyelination	Fed 0.2% CPZ in chow for 4 weeks.	*Mlkl^−/−^* mice have delayed demyelination in the caudal corpus callosum (MBP immunofluorescence).	Sex matched *Wt* and *Mlkl^−/−^* (CRISPR-Cas9) bred as separate cohorts. Same housing conditions since birth.	[68]
Parkinson’s disease (PD)	Intraperitoneal injection of MPTP.	*Mlkl^−/−^* show reduced neuroinflammatory markers (TNFα, IL-1β and IL-1 mRNA) and are significantly protected from striatal dopamine reduction and TH-positive neuron loss from the substantia nigra pars compacta compared to *Wt* controls.	Male *Mlkl^−/−^* (Jiahuai Han, Xiamen University, China) and *Wt* utilised.	[69]
Acute muscle injury	Intramuscular injection of cardiotoxin (CTX).	*Mlkl^−/−^* mice have fewer active muscle stem cells and thus suffer a significantly impaired capacity to regenerate muscle fibres. Regenerating myofibrils from *Mlkl^−/−^* mice express less myogenic factors MyoD, MyoG and nascent myofibril marker MYH3 in contrast to *Wt*.	Male *Mlkl^−/−^* (CRISPR-Cas9) and *Wt* mice utilised.	[70]
Hematological				
Venous thromboembolic (VTE) disease	100% flow obstruction of the IVC using ETHICON.	*Mlkl^−/−^* mice develop significantly smaller thrombi, with reduced areas of TUNEL+ cells, Ly6b+ neutrophils and F4/80+ macrophages. Systemically, *Mlkl^−/−^* mice have fewer circulating neutrophils, monocytes, and serum histone–DNA complexes following IVC ligation. No difference in baseline number of neutrophils, monocytes, DAMPs or bleeding time.	Male *Wt* (C57BL/6N) and *Mlkl^−/−^* [5].	[71]
Baseline	*Mlkl^−/−^* mice aged to 50 and 100 days.	*Mlkl^−/−^* mice are indistinguishable from *Wt* for the following blood parameters: WBC and lymphocyte count, thymic weight and cell count, splenic weight and cell count, lymph node weight and cell count.	Littermates derived from *Casp8^+/−^, Mlkl^+/−^* crosses [5] mice utilised.	[65]
Casp8 or FADD deficiency	Casp8 or FADD deficient background.	Compared with *Casp8^−/−^Ripk3^−/−^* or *Fadd^−/−^Ripk3^−/−^* mice, *Casp8^−/−^Mlkl^−/−^* or *Fadd^−/−^Mlkl^−/−^* demonstrate more severe lymphadenopathy and autoimmune manifestations.	Littermates derived from *Casp8^+/−^, Mlkl^+/−^* crosses [5] mice utilised.	[65]
Reconstitution of the hematopoietic system	Competitive transplantation assay using myeloablated recipients.	Following myeloablation, *Mlkl^−/−^* bone marrow stem cells were able to compete effectively with *Wt* counterparts for reconstitution of the hematopoietic system (blood, bone marrow and spleen).	Littermate controls.	[5]
*Mlkl^D139V^*-induced hematopoietic defects	Whole body homozygosity for constitutively active *Mlkl^D139V^* mutation.	P3 *Mlkl^D139V/D139V^* mice had significant deficits in lymphocyte and platelet counts in comparison to P3 *Mlkl^WT/WT^* and *Mlkl^D139V/WT^*.	Age and sex matched littermate controls.	[40]
	Myelosuppressive irradiation.	Recovery was delayed in *Mlkl^WT/D139V^* adult mice.	Age and sex matched littermate controls.	[40]
	5-fluorouracil.	Adult *Mlkl^D139V/WT^* mice had delayed recovery of hematopoietic stem and progenitor cells.	Age and sex matched littermate controls.	[40]
	Competitive bone marrow transplants.	Bone marrow-derived HSCs from *Mlkl^WT/D139V^* adults and foetal liver-derived HSCs from *Mlkl^WT/D139V^* and *Mlkl^D139V/D139V^* competed poorly or not at all with co-transplanted *Wt* bone marrow.	Age and sex matched littermate controls.	[40]
Acute thrombocytopaenia	Single dose of anti-platelet serum.	*Mlkl^−/−^* mounted a similar recovery to *Wt* in terms of magnitude and kinetics.	*Wt* and *Mlkl^−/−^* [5]	[72]
Hemostasis	Bleed times into 37 °C saline, after 3-mm tail amputations, measured over 10 min.	*Mlkl^−/−^* show prolonged bleeding times compared to *Wt* and yet equivalent total blood loss.	*Wt* and *Mlkl^−/−^* [5]	[72]
Reproductive system				
Male reproductive system aging	Male mice aged 15 months.	*Mlkl^−/−^* mice showed reduced body weight, reduced seminal vesicle weight, increased testosterone, increased fertility and fewer empty seminiferous tubules.	*Wt* and *Mlkl^−/−^* (CRISPR-Cas9) bred as separate cohorts. Same housing conditions since birth.	[73]
	Male mice aged 18 months.	No significant difference in mortality, body weight, testis weight, seminal vesicle weight or germ cell loss in seminiferous tubules between *Mlkl^−/−^* mice and *Wt* mice.	*Wt* and *Mlkl^−/−^* [5] are litter-mate controls.	[74]
Other				
Ventilator-induced lung injury (VILI)		*Mlkl^−/−^* mice were not protected against VILI at low or high tidal volumes.	Littermate *Wt* used for *Mlkl^−/−^* (Jiahuai Han Laboratory) mice.	[75]
Progressive renal fibrosis	Left urethral obstruction (UUO) by double ligation.	*Mlkl^−/−^* mice were not protected from renal damage.	Sex-matched male *Wt* and *Mlkl^−/−^* (Jiahuai Han Laboratory) mice.	[76]

* Methods not available.

## Data Availability

Not applicable.

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
