# Peer review of "The Role of the Key Effector of Necroptotic Cell Death, MLKL, in Mouse Models of Disease"

_biomolecules, 2021, doi:10.3390/biom11060803_

Round 1

Reviewer 1 Report

Review article submitted by Crutchfield and colleagues discusses the role of necroptosis effector protein MLKL in various mouse disease models. The authors define the major proteins that regulate necroptotic cell death and list (extensively) the studies where MLKL has been implicated in various disease through the use of animal models.

Overall, this review covers most of the relevant literature and discusses the role of MLKL in a comprehensive and systemic fashion. I am listing here a few relevant issues that should be addressed.

- In the abstract the authors claim that multi-billion dollar investments have been made in necroptosis blocking drugs. But it is not clear at all where they got that number and why is this, likely exaggerated, figure needed in this review?

- In the Introduction the authors state that necroptosis is characterized by the release of several DAMPs. However, it is important to clarify that the release of those DAMPs is by no means specific feature of necroptosis as this can happen during other forms of inflammation and cell death, for example pyroptosis.

- When discussing Neuro models and potential role of MKLK ascribed by various studies, the authors should carefully examine the literature describing the role of necroptosis regulators in neuro diseases for expression of MKLK in the brain and neuronal tissues. I was under impression that RIPK3 and MLKL are almost absent in these tissues and thus, the published data should be discussed with that perspective and caveat in mind.

Author Response

We thank the reviewers for their thoughtful comments.  We include here the reviewer’s comments ver batim, and our responses to these comments in blue italics. All major text changes to the manuscript are presented in red. 

Review article submitted by Crutchfield and colleagues discusses the role of necroptosis effector protein MLKL in various mouse disease models. The authors define the major proteins that regulate necroptotic cell death and list (extensively) the studies where MLKL has been implicated in various disease through the use of animal models.

 Overall, this review covers most of the relevant literature and discusses the role of MLKL in a comprehensive and systemic fashion. I am listing here a few relevant issues that should be addressed.

- In the abstract the authors claim that multi-billion dollar investments have been made in necroptosis blocking drugs. But it is not clear at all where they got that number and why is this, likely exaggerated, figure needed in this review?

Thanks, we agree that ‘multi-billion’ should be tempered. In Sheridan et al., 2019, a figure of >1.125 billion was given for the sale of rights to DNL747 and DNL758 by Denali Therapeutics to Sanofi. We arrived at multi-billion when factoring in the investment/license deals for even more advanced stages of GlaxoSmithKline RIPK1 inhibitor compounds but agree that the confirmed figure of >1billion is impressive enough, and gives the reader an important sense of the scale of interest in this therapeutic drug class. This now reads as follows:

‘This has triggered in excess of 1 billion-dollars (USD) in investment into the emerging class of necroptosis blocking drugs, and the potential utility of targeting the terminal effector is being closely scrutinised.’

- In the Introduction the authors state that necroptosis is characterized by the release of several DAMPs. However, it is important to clarify that the release of those DAMPs is by no means specific feature of necroptosis as this can happen during other forms of inflammation and cell death, for example pyroptosis.

We have edited this line to read as follows:

‘Like other forms of lytic cell death (e.g. pyroptosis),  necroptosis is characterised by the release of Damage-associated Molecular Patterns (DAMPs) and IL-33, IL-1α and IL-1b production as recently reviewed by [3].’

- When discussing Neuro models and potential role of MKLK ascribed by various studies, the authors should carefully examine the literature describing the role of necroptosis regulators in neuro diseases for expression of MKLK in the brain and neuronal tissues. I was under impression that RIPK3 and MLKL are almost absent in these tissues and thus, the published data should be discussed with that perspective and caveat in mind.

We completely agree and have edited the following line to further clarify this point to readers:

‘While opinions in the field remain split on the relative contribution of MLKL in haematological (which express high levels of MLKL) vs non-haematological (i.e. neurons express low levels of MLKL at baseline) cells in many of these models…’

Reviewer 2 Report

The review by Crutchfield et al. is interesting and well written. However, studies in mice must be compared with studies using human cells when possible, it will strengthen the relevance of the review since mice are unlikely to model the human diseases correctly.

The reviewer has also the feeling that some paragraphs are more developed than other. For instance, MLKL-/- mice are time to time compared with studies using RIPK3-/- mice but not always although the information is available. It is also the case in the context of Caspase-8 deficiency. Therefore, the authors need to harmonize the information throughout the manuscript.

The paragraph “metabolic disease” needs to discuss the recent findings showing that RIPK1 inhibition reduces obesity and fat deposition in liver most likely by reducing MLKL expression (PMID: 31760070). Also, ASO RIPK1-treated mice are protected against obesity (PMID: 32989316). It is very likely that direct pharmacological inhibition of MLKL (not yet available for mouse research) and ASO MLKL-treated mice will also reduce obesity and steatosis (https://pisa20.asip.org/PISA2020/assets/File/Abstract%20A073.pdf). For instance, in human cells in both primary hepatocytes and cell lines, the use of Necrosulfonamide (NSA) leads to a reduction of intrahepatic triglyceride (PMID: 31132314, PMID 31760070, PMID 32094147). Finally, Emricasan treatment worsened liver injury, hepatocyte ballooning and fibrosis in NASH patients (PMID: 31887369) by redirecting cells to alternative mechanisms of cell death, most likely necroptosis, further confirming a potential role of MLKL-dependent pathway in NAFLD. Therefore, the conclusion of this paragraph needs to be nuanced.

Author Response

We thank the reviewers for their thoughtful comments.  We include here the reviewer’s comments ver batim, and our responses to these comments in blue italics. All major text changes to the manuscript are presented in red.

The review by Crutchfield et al. is interesting and well written. However, studies in mice must be compared with studies using human cells when possible, it will strengthen the relevance of the review since mice are unlikely to model the human diseases correctly.

We thank the reviewer for this comment. While we agree this is a very important point, this would represent a very significant investment in work that is beyond the scope (and word limit) of this review. We hope that our title ‘The role of the key effector of necroptotic cell death, MLKL, in mouse models of disease’ makes the scope of our review clear to readers, and look forward to writing a comprehensive review on this very topic in the near future.

The reviewer has also the feeling that some paragraphs are more developed than other. For instance, MLKL-/- mice are time to time compared with studies using RIPK3-/- mice but not always although the information is available. It is also the case in the context of Caspase-8 deficiency. Therefore, the authors need to harmonize the information throughout the manuscript.

We agree that with such a wealth of animal model data available for both RIPK3 in many of these models, it may seem confusing that we do not have comprehensive coverage of results in RIPK3 KO mice. We have only mentioned RIPK3 mice when directly compared to MLKL within the same manuscript by the same groups (to ensure consistent methodology and thus accurate comparison), as RIPK3 KO mouse models are nicely covered by another review in the same issue of Biomolecules

Targeting RIP Kinases in Chronic Inflammatory Disease Mary Speir 1,2, Tirta M. Djajawi 1,2 , Stephanie A. Conos 1,2, Hazel Tye 1 and Kate E. Lawlor 1,2,*

The paragraph “metabolic disease” needs to discuss the recent findings showing that RIPK1 inhibition reduces obesity and fat deposition in liver most likely by reducing MLKL expression (PMID: 31760070). Also, ASO RIPK1-treated mice are protected against obesity (PMID: 32989316). It is very likely that direct pharmacological inhibition of MLKL (not yet available for mouse research) and ASO MLKL-treated mice will also reduce obesity and steatosis (https://pisa20.asip.org/PISA2020/assets/File/Abstract%20A073.pdf). For instance, in human cells in both primary hepatocytes and cell lines, the use of Necrosulfonamide (NSA) leads to a reduction of intrahepatic triglyceride (PMID: 31132314, PMID 31760070, PMID 32094147). Finally, Emricasan treatment worsened liver injury, hepatocyte ballooning and fibrosis in NASH patients (PMID: 31887369) by redirecting cells to alternative mechanisms of cell death, most likely necroptosis, further confirming a potential role of MLKL-dependent pathway in NAFLD. Therefore, the conclusion of this paragraph needs to be nuanced.

Thank you for your thoughtful suggestions, we agree there is so much more to be said about necroptosis and metabolism but are limited here due to our coverage of other disease classes. We have included the following new line to this section, but have not covered human cell lines and NSA for reasons mentioned earlier. We look forward to the publication of Li et al’s work on MLKL ASOs and obesity.

‘In line with these findings, mice treated with RIPA-56 (an inhibitor of RIPK1) downregulate MLKL expression and were found to be protected against HFD-induced steatosis [65]. These findings offer a tantalising clue that necroptosis may contribute to this disease, depending on the trigger. ’

Reviewer 3 Report

The presented work is a clearly written and well-researched review of the necrosome member mixed lineage kinase domain-like protein (MLKL) in the context of murine mouse models and diseases. The paper impresses with its excellent and up-to-date literature review and contrasts quite different seeming experimental results such as the dose-dependent effects of TNF in the SIRS model very well.

From a scientific point of view, however, I would be very pleased if the authors added their view on why inhibitors of necroptosis that are effective in murine mouse models fail in human trials. This aspect is briefly hinted at in the Abstract of the article, but perhaps it could be discussed in a little more detail - although it may of course be speculative - in the “Concluding remarks” section of the paper.

Minor comments to the authors:

  • On page 7, line 4, the literature source “Moerke et al. 2019” is sufficient (instead of all authors).

  • The sentence “This is of significance as 80 serotypes of S. pneumoniae are currently not covered in the adult vaccine” (on page 10) seems out of place and rather confusing in this position. I suggest deleting this sentence (including the associated reference) without replacement.

  • Also on page 10: “Of note, Mlkl-/- mice are indistinguishable from wild-type controls in CLP-induced polymicrobial shock (J. Wu et al., 2013)”. There is at least one paper (PMID: 32152555) in which the opposite is described (therein Figure S1). The authors should cite the latter study and include it also in Table 1.

  • In the chapter “Metabolic disease” (page 11), it should be explicitly mentioned that Mlkl-/- animals that were fed a high-fat diet for 4 months gained significantly less body weight than their wild-type littermates (PMID: 30837196).

  • Table 1 is fantastic, but the nomenclature (“hepatic IRI injury” and “renal IRI injury”) is partially incorrect. For example, “IRI” means “ischemia reperfusion injury”, so it is not correct to write “IRI injury” (which is equivalent to “ischemia reperfusion injury injury”).

Author Response

We thank the reviewers for their thoughtful comments. We include here the reviewer’s comments ver batim, and our responses to these comments in blue italics. All major text changes to the manuscript are presented in red.

The presented work is a clearly written and well-researched review of the necrosome member mixed lineage kinase domain-like protein (MLKL) in the context of murine mouse models and diseases. The paper impresses with its excellent and up-to-date literature review and contrasts quite different seeming experimental results such as the dose-dependent effects of TNF in the SIRS model very well.

From a scientific point of view, however, I would be very pleased if the authors added their view on why inhibitors of necroptosis that are effective in murine mouse models fail in human trials. This aspect is briefly hinted at in the Abstract of the article, but perhaps it could be discussed in a little more detail - although it may of course be speculative - in the “Concluding remarks” section of the paper.

Thank you for these comments and your suggestion. We have added the following line as part of the concluding remarks:

"This point is particularly poignant considering withdrawals of certain RIPK1 inhibitor drugs from Phase I (pancreatic cancer) and II (chronic inflammatory diseases) clinical trials due to lack of efficacy [91] [92]."

 Minor comments to the authors:

  • On page 7, line 4, the literature source “Moerke et al. 2019” is sufficient (instead of all authors).

This has been edited as suggested.

  • The sentence “This is of significance as 80 serotypes of S. pneumoniae are currently not covered in the adult vaccine” (on page 10) seems out of place and rather confusing in this position. I suggest deleting this sentence (including the associated reference) without replacement.

This has been removed

  • Also on page 10: “Of note, Mlkl-/- mice are indistinguishable from wild-type controls in CLP-induced polymicrobial shock (J. Wu et al., 2013)”. There is at least one paper (PMID: 32152555) in which the opposite is described (therein Figure S1). The authors should cite the latter study and include it also in Table 1.

Thank you for bringing this very important oversight to our attention. We have now added the findings of Chen et al 2020 to Table 1 and to the main body of the manuscript as follows:

‘Interestingly,  Mlkl-/- has been described to be both protective against [56], and dispensable in [6], the pathogenesis of  CLP-induced polymicrobial shock. Results derived from the CLP model of shock can be hard to replicate, given the inter-facility and even, intra-facility, heterogeneity of the cecal microbiota in mice [6].’

  • In the chapter “Metabolic disease” (page 11), it should be explicitly mentioned that Mlkl-/- animals that were fed a high-fat diet for 4 months gained significantly less body weight than their wild-type littermates (PMID: 30837196).

We have included the following line (page 12) as suggested:

‘Whilst comparable at baseline, after 16 weeks of HFD, Mlkl-/- mice gained significantly less body weight, notably visceral adipose tissue, than their wild-type littermates [67].’ 

  • Table 1 is fantastic, but the nomenclature (“hepatic IRI injury” and “renal IRI injury”) is partially incorrect. For example, “IRI” means “ischemia reperfusion injury”, so it is not correct to write “IRI injury” (which is equivalent to “ischemia reperfusion injury injury”).

Thank you for pointing this out, we have carefully replaced all occurrences of ‘IRI injury’ with ‘IRI’ alone

Reviewer 4 Report

In this review, the authors summarized dozens of previous reports testing MLKL’s contribution to mouse models of disease and discussed the pathological role of the MLKL.  This review is concisely well-written. It reads well. I highly appreciate their efforts to carefully read an enormous number of the papers and accurately describe experimental designs and results. Obviously, as the authors point out, preparing proper control is critical in any type of mouse experiments. This review not only is informative but also will give a caution to readers about interpretation of results obtained from experiments without proper control. I do not have any concern about this manuscript and do recommend to publish as it is.

Author Response

We thank the reviewers for their thoughtful comments.  We include here the reviewer’s comments ver batim, and our responses to these comments in blue italics. All major text changes to the manuscript are presented in red.

In this review, the authors summarized dozens of previous reports testing MLKL’s contribution to mouse models of disease and discussed the pathological role of the MLKL.  This review is concisely well-written. It reads well. I highly appreciate their efforts to carefully read an enormous number of the papers and accurately describe experimental designs and results. Obviously, as the authors point out, preparing proper control is critical in any type of mouse experiments. This review not only is informative but also will give a caution to readers about interpretation of results obtained from experiments without proper control. I do not have any concern about this manuscript and do recommend to publish as it is.

We thank the reviewer for their feedback.

Round 2

Reviewer 2 Report

this is a really nice, comprehensive and constructive review article, authors have responded to my comments accordingly.